# CLaMR: Contextualized Late-Interaction for Multimodal Content Retrieval

## Abstract

Online video content is richly multimodal: a single video might blend vision, speech, ambient audio, and on-screen text. Conventional retrieval systems typically treat these modalities as independent retrieval sources, which can lead to noisy and subpar results. In this work, we explore multimodal video content retrieval, where relevance can be scored from a single modality or jointly across multiple modalities. Consequently, an effective retriever must dynamically determine which modality (or set of modalities) best address a given query. We introduce CLaMR, a multimodal, late-interaction retriever that jointly indexes four modalities: video frames, transcribed speech, on-screen text, and metadata. CLaMR jointly encodes all modalities within a unified multimodal backbone for improved contextualization and is trained to enhance dynamic modality selection via two key innovations. First, to overcome the lack of suitable training data, we introduce MultiVENT 2.0++, a large-scale synthetic dataset built on MultiVENT 2.0 (a collection of event-centric videos in various languages paired with English queries) with modality-targeted queries to teach modality selection. Next, we propose a modality-aware contrastive loss that trains the model on both a standard contrastive objective and an objective for learning correct modality usage. On the test sets of MultiVENT 2.0++ and MSRVTT, we observe that conventional aggregation strategies, such as averaging similarities for baseline retrievers, often degrade performance by introducing noise from irrelevant modalities. In contrast, CLaMR consistently outperforms existing retrievers: on MultiVENT 2.0++, CLaMR improves nDCG@10 by 25.6 points over the best-performing single-modality retriever and by 35.4 points over the best-performing multi-modality retriever. We illustrate the downstream utility of CLaMR with experiments on long-video QA, where it improves performance by 3.50% over LanguageBind on Video-MME and 1.42% over dense frame sampling on LongVideoBench.[1]

## 1 Introduction

Online platforms host a massive stream of video content that is natively *multimodal*, intertwining visual scenes, spoken dialogue, ambient sound, on-screen text, and free-form descriptions (Samuel et al., 2025). Modern search engines and retrieval-augmented generation (RAG) systems must therefore decide, for every user query, *which* of these heterogeneous sources actually contains useful data and *how* to exploit it (Cho et al., 2024). However, effectively searching this rich content requires combining signals from diverse sources in ways that prior work has not fully addressed. Existing approaches often focus on a single modality (e.g., video) or convert content to text via captioning or OCR (Memon et al., 2020; Smith, 2007), which risks missing key information encoded in the original modality (Cho et al., 2024; Faysse et al., 2025). Furthermore, multimodal search engines that do treat different modalities as separate sources often rely on simple heuristics for merging scores, such as maximum or reciprocal-rank fusion (RRF) (Cormack et al., 2009), as illustrated in Figure 1. These methods implicitly assume that multiple modalities will agree on relevance but risk drowning out valuable evidence from one modality with noise from another. In fact, as we show in Table 1, simple combination methods like averaging often lead to worse performance than using the best single modality, primarily due to limited cross-modal understanding.

---

[1]Code and data are in the supplementary materials.

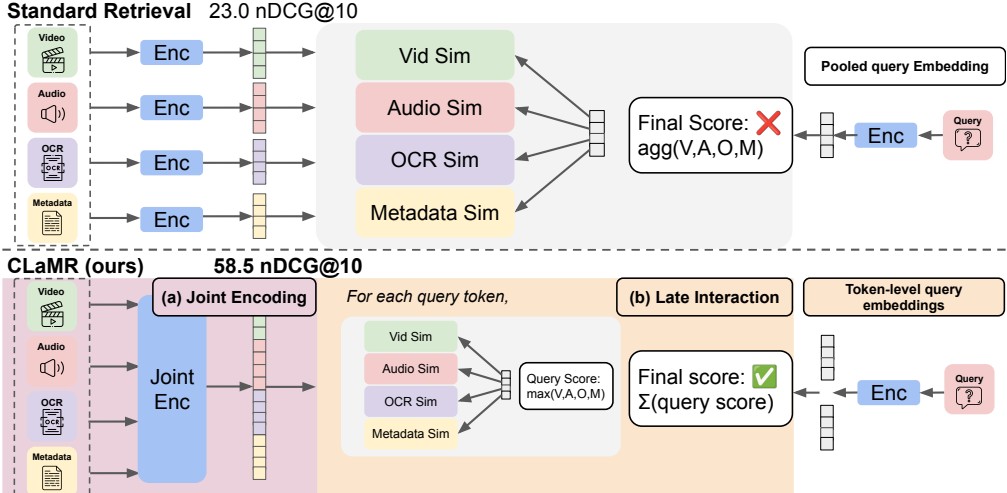

Figure 1: Illustration of the multimodal video content retrieval task. A query derived from a video's audio is presented. Conventional retrieval systems (top) encode each modality independently and then aggregate their similarity scores, a process easily contaminated by noise from irrelevant modalities. By contrast, CLAMR (bottom) jointly encodes all modalities (a) and, via a late-interaction mechanism (b), computes fine-grained, token-level similarities that dynamically focus on the most relevant modalities (audio and video) for the query.

To close this gap, we introduce CLAMR, a contextualized, late-interaction retriever that jointly encodes video frames, speech transcripts, on-screen text, and other metadata. Originally studied in text retrieval, late-interaction (LI) models first independently encode queries and documents, then compute lightweight but fine-grained token-level similarities to facilitate precise relevance judgments (Khattab & Zaharia, 2020; Santhanam et al., 2022). This contrasts with standard bi-encoder retrievers that compute only a single cosine similarity between pooled query and document embeddings (Fig. 1, top). While promising, late interaction has primarily been studied in text-based contexts, with its multimodal application largely restricted to single modalities like images (Faysse et al., 2025) or video frames (Reddy et al., 2025). Meanwhile, applying late interaction to retrieving multimodal video content has remained unexplored. Inspired by recent vision-language models that jointly process cross-modal inputs (Chen et al., 2023b;c; Sun et al., 2024), we propose to address this gap by using a single vision-language backbone to encourage better contextualization. As shown in Fig. 1 (bottom), by encoding *all sources together* rather than in isolation, CLAMR learns directly from contrastive signals which modality to trust for each query, eliminating the need for fragile combination techniques or routers (Yeo et al., 2025) that require extra computation. To teach CLAMR to retrieve the correct content and focus on the correct modality, we propose a modality-aware contrastive loss (Sec. 3.3). Our loss explicitly encourages CLAMR to assign the highest similarity score to modalities containing query-relevant information. For example, for a query derived from speech, the model should learn to match evidence from the audio signal over other modalities (Fig. 2).

Finally, to effectively train a late-interaction retriever that can dynamically select among modalities, we introduce MULTIVENT 2.0++ (Sec. 4), a large-scale synthetic dataset built upon MULTIVENT 2.0 (Kriz et al., 2025). While MULTIVENT 2.0 provides a massive set of multimodal data, it lacks sufficient modality-specific queries for training. MULTIVENT 2.0++ addresses this by synthesizing 371k queries specifically targeting different modalities for the unannotated videos.

On MULTIVENT 2.0++ and the popular text-video retrieval benchmark MSR-VTT (Xu et al., 2016), CLAMR substantially outperforms all unimodal and multimodal baselines. For example, on MULTIVENT 2.0++, CLAMR surpasses the strongest unimodal and multimodal baselines by 25.7% nDCG@10. Our ablation studies highlight the critical roles of contextualization and modality-aware contrastive training. We also demonstrate the downstream benefits of CLAMR on long-video question answering, where CLAMR retrieves relevant segments given a query. With a fixed frame budget, CLAMR provides improvements over LanguageBind on both VideoMME (Fu et al., 2024) and LongVideoBench (Wu et al., 2024) by retrieving more relevant video segments.

## 2 RELATED WORK

**Multimodal Retrievers.** Multimodal retrievers aim to align and retrieve information across different modalities such as text, image, audio, and video. A key development is large-scale vision-language pretraining with contrastive learning, as exemplified by dual-encoder models like CLIP (Radford et al., 2021) and ALIGN (Jia et al., 2021). These models learn joint embedding spaces for images and text, inspiring extensions to additional modalities. For instance, ImageBind (Girdhar et al., 2023) extends contrastive alignment to audio and other inputs, while LanguageBind (Zhu et al., 2024) uses language to bind video and diverse modalities. Recent retrievers also incorporate signals such as OCR-extracted text (Zhang et al., 2024), speech transcripts (ASR), and video frame features (Reddy et al., 2025). However, dynamically selecting the most relevant modality for each query remains a challenge, as most systems fuse modalities in a fixed way. Emerging benchmarks like MULTIVENT (Kriz et al., 2025) emphasize this challenge by providing queries that require retrieval from whichever modality contains the answer, underscoring the need for adaptive retrievers. Our work addresses this gap by training a single retriever to dynamically identify and focus on the most relevant modality per query. Recent system for multimodal document retrieval (Zhan et al., 2025) often fuses (concatenates) modalities into a unified single embedding for retrieval. Our approach is fundamentally different: we employ a late-interaction framework that preserves fine-grained, token-level representations across all modalities. This architecture allows our model to dynamically select the most relevant modality for any given query, a capability we foster through a novel, modality-aware training objective and our co-designed MULTIVENT 2.0++ dataset.

**Late Interaction.** Unlike standard dual encoders that match queries and documents via coarse-grained similarity (Karpukhin et al., 2020; Reimers & Gurevych, 2019), or cross-encoders that compute full query-document interactions at a high computational cost (Wang et al., 2020), late-interaction methods offer a middle ground. They enable fine-grained token-level matching while retaining much of the efficiency of dual encoders. ColBERT (Khattab & Zaharia, 2020) introduced this paradigm for text, and ColBERTv2 (Santhanam et al., 2022) improved its effectiveness. Originally developed for monolingual text, late interaction has been extended to new languages and modalities. For instance, JaColBERTv2.5 (Clavié, 2024) explored multilingual retrievers, while ColPali (Faysse et al., 2025) applied a ColBERT-style model to document images. These approaches allow token-level comparisons across modalities, e.g., matching a query word to a specific image region. Notably, video retrieval methods like CLIP4Clip (Luo et al., 2021) still rely on pooled global embeddings, whereas late-interaction models preserve multiple embeddings per item for detailed matching. Our approach differs by introducing modality-wise late interaction that computes token-level scores separately across modalities and trains the model to select the most relevant one dynamically.

## 3 CLAMR

We propose CLAMR (**C**ontextualized **La**te-interaction for **M**ultimodal content **R**etrieval), a novel retrieval framework capable of attending to different views of multimodal content (e.g., frames, speech, metadata). Unlike previous methods that encode each modality separately, CLAMR focuses on contextualization by encoding all modalities together and employs late interaction for fine-grained retrieval. Below, we detail the task setup, architecture, and training objective.

### 3.1 TASK SETUP

Given a query $q$, the retriever must identify the most relevant document $d$. Each document $d = \{v, a, o, m, \dots\}$ may contain *multiple* modalities, such as video $v$, audio $a$, on-screen text $o$, and metadata $m$. An example is depicted in Fig. 2. The core challenge is to locate the relevant document, as the evidence might be found within a single modality or distributed across several.

### 3.2 CONTEXTUALIZED MULTIMODAL ENCODER

We primarily employ a vision-language model (VLM), which is essential for leveraging detailed token- and patch-level interactions because it jointly encodes all modalities. As illustrated in Fig. 2, all input modalities are first concatenated into a single sequence, with visual inputs preceding textual inputs. The VLM then processes this combined sequence to generate contextualized hidden states

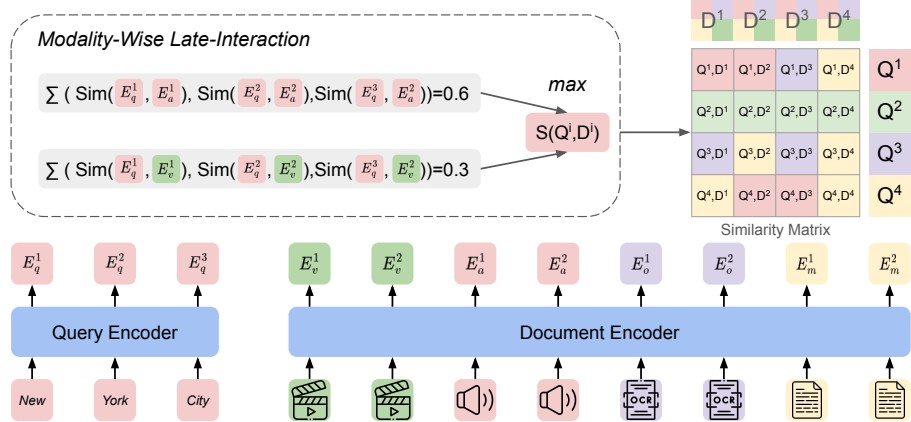

Figure 2: CLAMR with modality-wise late interaction for multimodal contrastive learning. A text query and a multimodal video document (comprising visual, audio, OCR, and metadata signals) are encoded by the model. Token-level late interaction yields a similarity score for each modality; the highest of these scores becomes the query-document similarity. Similarities for the positive pair and in-batch negatives are then fed to a contrastive loss.

for all tokens and patches. Finally, these hidden states are passed through a projection layer to produce the final representation for each token. See Sec. 5 for more details.

**Omni-Models.** Since ASR transcripts are converted to text for VLMs, we also explore integrating CLAMR with omni-models that can process audio directly. Unlike VLMs, omni-models such as Qwen-Omni (Xu et al., 2025) can process raw audio. The setup generally follows that of the VLM, with the exception of using raw audio instead of ASR transcripts.

### 3.3 CONTEXTUALIZED LATE-INTERACTION.

All hidden states are projected into a shared embedding space. A query yields $\mathbf{E}_q \in \mathbb{R}^{N_q \times D}$, where $N_q$ is the number of query tokens. Each document is represented by concatenating the embeddings of all its modalities, yielding a single representation $\mathbf{E}_d \in \mathbb{R}^{N_d \times D}$, where $N_d = \sum_{m \in \mathcal{M}} N_{d,m}$ is the total number of tokens across all modalities. Late interaction (LI) (Khattab & Zaharia, 2020; Santhanam et al., 2022; Faysse et al., 2025) compares *token-level embeddings*: for each query token, its maximum cosine similarity to any document token is computed, and these maximum similarities are then summed over all query tokens. At inference time, we use this contextualized scoring to allow each query token to match the most similar document token from *any modality*:

$$\mathrm{LI}_{\mathrm{context}}(q, d) = \sum_{i=1}^{N_q} \max_{j=1}^{N_d} \langle \mathbf{E}_q^{(i)}, \mathbf{E}_d^{(j)} \rangle. \tag{1}$$

### 3.4 TRAINING OBJECTIVE: MULTIMODAL CONTRASTIVE LEARNING.

Our goal is to train the model to not only retrieve the correct document but also dynamically select the optimal modality. Let $\{(q_k, d_k)\}_{k=1}^{b}$ be a batch with one query per document. We use the standard InfoNCE loss (van den Oord et al., 2019) to train the model to retrieve the correct document from a batch of negatives. As illustrated in Figure 2, the loss is formulated as follows:

$$\mathcal{L}_{\mathrm{InfoNCE}} = -\frac{1}{b} \sum_{k=1}^{b} \log \frac{\exp(s_{k,k}/\tau)}{\sum_{j=1}^{b} \exp(s_{k,j}/\tau)}, \tag{2}$$

where $\tau$ is temperature, and $s_{i,j}$ is the similarity score between query $q_i$ and document $d_j$.

**Multimodal Video Content**

| 📽️ **Video Title:** | 🔊 **ASR:** | 🔤 **OCR:** | 📄 **Metadata:** |
|---|---|---|---|
| The Science of Soil Health: Nature's Way … | In our series on the science of soil health… | Origins of root-mediated pH char soil… | Just when you thought soil microbes … |

Prompt a LLM to generate one modality-specific query for each modality

| **Query (base):** | **Query (speech):** | **Query (OCR):** | **Query(description):** |
|---|---|---|---|
| *Soil health mineral extraction video* | *CO2 role in soil mineral weathering* | *PH changes in soil biogeochemistry* | *Soil microbes providing carbon dioxide to plants* |

Figure 3: Illustration of deriving modality-specific queries from multimodal video content. An LLM uses the title, ASR, OCR, and metadata separately to generate queries that can be answered *primarily* by the designated modality.

**Modality-Wise Late-Interaction.** While contextualized late interaction could be used directly for the similarity score $s_{i,j}$, we find that the model struggles to learn modality usage effectively, as it must simultaneously differentiate between examples and between modalities within the same example. Thus, we employ a more factorized formulation *during training*. Here, we separately compute the late-interaction score for each modality and select the maximum score. Since the queries in MULTIVENT 2.0++ are designed to target a single modality, this approach guides the model to attend to one modality at a time, allowing it to focus on differentiating between distinct examples. The similarity is defined as:

$$\text{LI}_{\text{mw}}(q, d) = \max_{m \in \mathcal{M}} \sum_{i=1}^{N_q} \max_{j=1}^{N_{d,m}} \langle \mathbf{E}_q^{(i)}, \mathbf{E}_{d,m}^{(j)} \rangle. \tag{3}$$

As shown in Fig. 2, after computing per-modality scores between an audio query and the document's modalities, the score from the audio modality is the highest; this score is used as the final similarity for that pair. These scores for all pairs in the batch form a similarity matrix where the diagonal represents positive pairs and off-diagonals represent negative pairs.

## 4 MULTIVENT 2.0++: AUGMENTING DATA FOR MULTIMODAL RETRIEVAL

To train a retriever to actively decide which modality to focus on, the training set must include queries that are unambiguously grounded in a single modality. MULTIVENT 2.0, however, was not designed with this goal in mind: most of its 101K videos lack any queries, and the obvious fallback—using the video title as a query—yields short, generic prompts that neither single out a modality nor, in many cases, even appear in English. Among the 10K videos that *are* annotated, only 1,504 queries are provided, a number too small to adequately train retrievers for fine-grained modality selection. To address this limitation, we introduce MULTIVENT 2.0++, augmenting training queries for MULTIVENT 2.0 on the unannotated videos.

**Synthetic Expansion of Modality-Specific Queries.** Building on the design of the original annotations—where each annotated video includes a 'base' query plus one specific query each for audio, OCR, and metadata—we automatically extend this schema to 91k unannotated videos. For each unannotated video, we first collect its modality sources: ASR transcripts, frame-level OCR text, and video metadata (comprising title and human-written description). Subsequently, for each modality source, we construct an in-context prompt consisting of ten human-written, modality-specific query-content pairs randomly sampled from our annotated corpus. A large language model (LLM) is then prompted with these examples to generate a base query (loosely derived from the video title) and one new modality-specific query for each of these sources. The LLM is instructed to phrase these generated queries such that a correct answer can be retrieved *primarily* from the respective target modality. Fig. 3 shows this generation pipeline. Our approach allows the LLM to generate queries whose answers may occasionally be present in more than one modality—for instance, the term `pH change` might appear in both OCR and ASR—thus encouraging the retriever to weigh corroborating evidence rather than enforcing an artificially one-to-one query-modality mapping.

**LLM Choice for Synthetic Data Generation.** Because many videos contain non-English text, the generator must both translate and condense content. We therefore use *Gemma-3-27b-it* (Team et al., 2025), whose strong multilingual abilities make it well-suited to producing English queries from diverse source languages. Furthermore, this model has demonstrated strong performance in various NLP tasks, making it an appropriate choice for generating high-quality queries.

**Dataset Split.** Our training set consists of all synthetically generated queries and their associated document, totaling 371,644 query-document pairs. From this set, we allocate 367,644 pairs for training and 4,000 pairs for our validation set. For testing, we utilize the public benchmark split of MULTIVENT 2.0, which comprises 1,504 queries with available human judgments, as its private benchmark split does not provide these. Importantly, the videos corresponding to these 1,504 MUL-TIVENT 2.0 test queries were human written and were not used in the generation process of our synthetic data generation to ensure the model is not learning from data artifacts.

**Quality Assurance.** To ensure quality of the generated queries, our prompt includes in-context, human-annotated examples. We then validated query quality using both manual and automatic methods. In a manual review of 400 random queries, two of the authors independently assessed whether each query was relevant and logically derivable from its source modality, and provided a binary judgment of its relevance. They confirmed a high query relevance rate of 86.41%. A larger-scale automatic validation on 4,000 examples, where a model (Gemma-3-27b-it) was prompted to rate query-source relevance on a 1-5 scale, yielded a similarly strong average score of 4.22 out of 5. Both assessments confirm the high quality of our generated queries, with further details provided in Appendix B.4.

## 5 EXPERIMENTAL SETUP

**CLAMR Implementation Details.** We use `Qwen-VL-2.5-3B`[2] (Bai et al., 2025) as the backbone for CLAMR with VLM, which offers strong multimodal accuracy at a modest size. For the Omni-model variant, we experimented with `Qwen-Omni-3B`[3](Xu et al., 2025), which utilizes Whisper (Radford et al., 2022b) as its underlying audio encoder. We append a 128-dimensional linear projection layer, following ColPali (Faysse et al., 2025). We train separate versions of CLAMR on MULTIVENT 2.0++ and MSRVTT for 1 and 5 epochs, respectively. Training is performed using a batch size of 16, distributed across 8 A100 80GB GPUs.To reduce memory usage, we employ 4-bit quantization with QLoRA (Dettmers et al., 2023), setting the LoRA rank $r = 128$ and $\alpha = 128$. Our implementation is built on the *transformers* library (Wolf et al., 2020). Unless noted otherwise, we keep default hyperparameters, train with the 8-bit Adam optimizer, and set the learning rate to $1 \times 10^{-5}$ for all experiments. Training on MULTIVENT 2.0++ required approximately 10 hours, while training on MSRVTT took about 4 hours. More details can be found in Appendix A.

**Baselines.** As single-modality baselines, we use multilingual CLIP (mCLIP)[4] from Reimers & Gurevych (2019) by processing only their corresponding modality (video, audio, OCR, or metadata). For multi-modality baselines, we use several strong encoders: ImageBind (Girdhar et al., 2023), and LanguageBind (Zhu et al., 2024). For the two models, we average the similarity scores obtained from all available modalities, a method we found to yield the best performance with these models. We also include results using a router as an aggregation method. For this approach, we utilize GPT 4.1 to predict the most relevant modality given the query and then use the similarity score from that predicted modality as the final similarity score. Finally, as an additional strong baseline, we fine-tune the Qwen-VL 2.5 backbone (the same used for CLAMR) with a standard contrastive loss. This involves using the embedding of the last token as the pooled representation for a sequence, a common practice in VLM fine-tuning (Bao et al., 2022; Ouali et al., 2025; Jiang et al., 2025).

**Datasets.** Our primary evaluation dataset is MULTIVENT 2.0++, where we train on our synthetically generated data and evaluate on the original public evaluation from MULTIVENT 2.0. This testing setup consists of 1,504 query-document pairs. We also include MSR-VTT (Xu et al., 2016),

---

[2]https://huggingface.co/Qwen/Qwen2.5-VL-3B-Instruct
[3]https://huggingface.co/Qwen/Qwen2.5-Omni-3B
[4]https://huggingface.co/sentence-transformers/clip-ViT-B-32-multilingual-v1

Table 1: Retrieval results on MULTIVENT 2.0++ and MSRVTT. * indicates statistical significance ($p < 0.05$) compared to other baseline methods with a paired bootstrap test.

| Method | Modality | MULTIVENT 2.0++ | | | | MSR-VTT | | | |
|---|---|---|---|---|---|---|---|---|---|
| | | R@1 | R@5 | R@10 | nDCG@10 | R@1 | R@5 | R@10 | nDCG@10 |
| *Single-Modality* | | | | | | | | | |
| ICDAR + mCLIP | OCR | 2.9 | 10.4 | 14.7 | 8.1 | - | - | - | - |
| Whisper + mCLIP | Audio | 4.5 | 19.7 | 24.5 | 13.9 | 5.2 | 8.7 | 10.8 | 7.7 |
| Description + mCLIP | Metadata | 7.5 | 24.9 | 29.5 | 18.1 | - | - | - | - |
| Video + mCLIP | Vision | 10.1 | 35.9 | 45.7 | 26.8 | 27.1 | 50.6 | 61.6 | 42.7 |
| Imagebind | Vision | 15.4 | 43.0 | 52.1 | 32.8 | 28.9 | 52.8 | 63.3 | 44.9 |
| LanguageBind | Vision | 14.2 | 39.5 | 47.9 | 30.2 | 40.2 | 64.3 | 74.8 | 56.5 |
| *Multi-Modality* | | | | | | | | | |
| mCLIP (avg.) | All | 7.9 | 31.9 | 39.7 | 23.0 | 19.5 | 38.3 | 47.0 | 32.2 |
| mCLIP (router) | All | 7.0 | 29.0 | 34.8 | 20.5 | - | - | - | - |
| ImageBind (avg.) | All | 3.9 | 10.6 | 14.0 | 8.5 | 20.4 | 35.7 | 43.0 | 30.9 |
| ImageBind (router) | All | 8.9 | 22.2 | 27.3 | 17.7 | - | - | - | - |
| LanguageBind (avg.) | All | 6.8 | 19.8 | 23.7 | 15.1 | 23.0 | 38.3 | 45.2 | 33.2 |
| LanguageBind (router) | All | 9.8 | 27.3 | 33.2 | 21.0 | - | - | - | - |
| Qwen VL 2.5 pooled | All | 21.6 | 74.8 | 81.6 | 52.2 | 36.2 | 62.9 | 73.9 | 53.8 |
| *Ours* | | | | | | | | | |
| CLAMR (Omni) | All | 25.5 | 81.1 | 85.2 | 55.7 | 45.5 | 69.8 | **81.0** | 62.1 |
| CLAMR (VLM) | All | **26.7*** | **85.1*** | **88.0*** | **58.5*** | **46.1*** | **71.3*** | 79.8* | **62.4*** |

a standard text-video retrieval benchmark used in several prior works (Zhu et al., 2024; Chen et al., 2023a;b). Following prior work (Luo et al., 2021; Chen et al., 2023b), we split the 10K examples of MSRVTT into 9K and 1K, for training and evaluation, respectively. We also conducted experiments on additional standard video-retrieval datasets in Appendix B.2.

**Metrics.** Following standard practice in retrieval evaluation (Liu, 2009; Thakur et al., 2021), we evaluate the models performance using standard retrieval metrics: **Recall@k and nDCG@10** (Järvelin & Kekäläinen, 2002). Recall@k measures whether a relevant item appears in the top-$k$ retrieved results, while normalized Discounted Cumulative Gain (nDCG) accounts for both the relevance and rank of retrieved items, assigning higher scores when highly relevant items appear early in the ranked list and penalizing relevant items that appear lower. We use the top-10 cutoff (nDCG@10) to balance sensitivity and efficiency in ranking evaluation.

## 6 RESULTS

We present our main retrieval results, ablation studies, and a downstream application in long-video QA. We provide further analysis in Appendix B, including experiments an alternative contrastive loss formulations, per-modality performance breakdowns, results on additional video-text benchmarks, an efficiency analysis, and ablations on missing modalities. In Appendix C, we also provide additional qualitative examples that provide interpretability of how late-interaction selects different modalties to provide corroborative benefits.

### 6.1 RETRIEVAL RESULTS

The results, presented in Tab. 1, demonstrate that CLAMR (VLM) consistently outperforms both single-modality and multimodal baselines across all standard evaluation metrics. A key observation is the challenge faced by conventional multimodal baselines when attempting to fuse information from various modalities. For instance, models like mCLIP, ImageBind, and LanguageBind often exhibit diminished performance compared to their vision-only versions when using an average merging strategy (avg.) for all modalities. On MSR-VTT, LanguageBind (Vision; the best performing single-modality baseline model) achieves an R@1 of $40.2\%$, while its multimodal average (LanguageBind avg.) scores only $23.0\%$. This trend is also evident on MULTIVENT 2.0, where ImageBind (Vision) reaches $15.4\%$ R@1, substantially higher than the $3.9\%$ from ImageBind (avg.). This suggests that naive fusion methods are susceptible to noise or suboptimal integration of complementary information from diverse modalities, thereby hindering overall retrieval accuracy. Interestingly, employing

Table 2: Ablation study on MULTIVENT 2.0. B-C: impact of architectural and objective choices. D-H: CLAMR trained and tested on a single modality.

| | Method | Inference modality | R@1 | R@5 | R@10 | nDCG@10 |
|---|---|---|---|---|---|---|
| (A) | CLAMR | All | **26.66** | **85.11** | **88.03** | **58.47** |
| | *Architecture and training objective design* | | | | | |
| (B) | CLAMR without contextualization | All | 18.95 | 64.30 | 68.02 | 44.53 |
| (C) | CLAMR with $LI_{context}$ (instead of $LI_{mw}$) | All | 23.93 | 80.92 | 86.04 | 56.26 |
| | *Single-modality* | | | | | |
| (D) | CLAMR Base | Vision | 16.22 | 57.58 | 65.49 | 40.71 |
| (F) | CLAMR Audio | Audio | 18.15 | 64.56 | 68.48 | 43.93 |
| (G) | CLAMR OCR | OCR | 19.68 | 62.10 | 67.95 | 43.19 |
| (H) | CLAMR Metadata | Metadata | 20.01 | 68.22 | 72.94 | 47.09 |

a router strategy for these multimodal baselines on MULTIVENT 2.0++ shows a notable improvement over the average merging strategy, though still falling short of vision-only performance in some cases. For example, LanguageBind (router) shows a marked improvement with an R@1 of 9.8% compared to LanguageBind (avg.) at 6.8%, but remains lower than LanguageBind (Vision) at 14.2%. This indicates that while routing can be more effective than simple averaging for multimodal fusion, it does not consistently outperform the strongest single-modality inputs for these baselines.

In stark contrast, CLAMR demonstrates superior performance by effectively leveraging multimodal information. The VLM variant achieves the highest scores across all reported metrics on both datasets except for R@10 on MSR-VTT where the Omni-model variant outperforms the VLM variant. On MSR-VTT, CLAMR achieves an R@1 of 46.1%, surpassing the strongest single-modality baseline (LanguageBind Vision) by 5.9%. The performance gains are even more pronounced on the MULTIVENT 2.0 dataset, where the queries target different modalities. Here, CLAMR achieves an R@1 of 26.7%, which is 11.3% higher than the best performing single-modality baseline. These results underscore the efficacy of our proposed approach in robustly integrating multimodal signals for enhanced retrieval. The VLM demonstrates superior overall performance compared to the Omni-model, particularly on MULTIVENT 2.0++. We hypothesize this advantage stems from the Omni-model's architecture: accommodating speech tokens reduces its capacity for handling extended sequence lengths, and in turn restricts batch sizes, impairing the effectiveness of contrastive learning. Consequently, we focus primarily on the VLM for our subsequent results.

## 6.2 ABLATION STUDIES

To understand the contributions of different components of our proposed CLAMR architecture and training strategy, we report ablation studies on the test set of MULTIVENT 2.0++ in Table 2.

**Impact of Contextualization.** First, we investigate the impact of contextualization, where we jointly encode all the modalities in a single pass to the model. By removing the contextualization mechanism from our full model (B), where we encode each modality separately and then concatenate all the representations back together, we observed a substantial decrease in performance across all metrics. Specifically, R@10 by 20.01% and nDCG@10 by 13.94%, highlighting contextualization's critical role in effectively fusing information from multiple modalities for improved retrieval.

**Impact of Late-interaction.** Next, we compare our proposed training objective with the contextualized late-interaction ($LI_{context}$) (C). While the $LI_{context}$ model performs competently, our full model (A) achieves superior results with an improvement of 1.99 in R@10 and 2.21 in nDCG@10 compared to model (C). This suggests that our training objective facilitates a more effective learning process for the model, enabling better integration and utilization of multimodal signals.

**Comparing Joint and Unimodal Training.** We compare the performance of models trained with only a single modality to our full model trained with all modalities. When trained and evaluated on their respective single modalities, these models performed considerably worse than the full multimodal model. For instance, the the base model (D) achieves an nDCG@10 of only 40.71, a substantial drop of 17.76 points from the full model's score. Among these unimodal variants, metadata

Table 3: Modality Accuracy on modality-specific setting.

|  | Vision | Audio | OCR | Metadata | Avg. |
|---|---|---|---|---|---|
| Router | 22.4 | 30.9 | 20.0 | 56.9 | 30.9 |
| mCLIP - max | 0.0 | 54.3 | 69.1 | 39.2 | 39.5 |
| CLAMR | **58.2** | **80.0** | **84.3** | **86.0** | **76.4** |

proves to be the most informative single source. The base model that uses vision performs the worst, most likely because the training base queries are generated using video titles. This large performance gap demonstrates that training the model with comprehensive multimodal information leads to the most robust and effective retrieval system.

## 6.3 QUERY-SPECIFIC ANALYSIS

To better understand whether CLAMR correctly identifies and retrieves from the intended modality, we conduct a fine-grained evaluation under modality-specific settings. This section describes how we construct and validate modality-targeted queries, and how we use them to evaluate retrieval accuracy.

**Filtering Human-Written Queries.** We begin with a small pool of human-annotated queries from MULTIVENT 2.0 and apply an LLM-based filtering pipeline to verify their modality specificity. For each query, we prompt the model to judge whether the answer is uniquely grounded in the annotated target modality or also available in other modalities. A query passes this filter only if it is judged answerable solely from the intended modality. For example, to assess video-grounded queries, we use Qwen2.5-VL-72B-Inst to caption the visual content and check whether other textual modalities (ASR, OCR, metadata) could also provide the answer. This filtering process yields a small but verified set of modality-pure queries, which we use for preliminary analysis.

**Generating Synthetic Modality-Specific Queries.** To scale this analysis, we generate new queries using an LLM prompted with four modality-specific documents (video, ASR, OCR, and metadata) and instructed to produce a query answerable only by one target modality. We then reapply our filtering step to verify that no other modality could answer the generated query. The surviving examples are passed to human annotators for final verification. This expanded dataset allows us to compute modality-specific retrieval accuracy at a larger scale.

**Results and Accuracy Breakdown.** We use this filtered dataset to evaluate whether a retriever correctly attends to the intended modality when answering modality-specific queries. In Table 3, we report modality-wise accuracy for CLAMR and a strong routing baseline. The router selects a modality per query based on similarity to query type embeddings and executes retrieval only within that modality, and for mCLIP we use the modality that scores the highest similarity.

CLAMR dramatically outperforms the router and mCLIP baseline across all modalities, achieving an average accuracy of 76.4% versus 30.9%. Notably, it achieves particularly high accuracy for OCR (84.3%) and ASR (80.0%), confirming that it learns to focus on the correct modality without explicit routing. In contrast, the router fails to adapt to the content of the query and performs poorly on modalities like video and OCR and mCLIP fails to make use of the video modality. These results validate that our training objective and architecture enable effective query-specific modality selection, without the need for fragile routing heuristics.

Furthermore, the accuracy for Video is lower than the other modalities. This is similar to the trend of CLAMR's query retrieval performance, as shown in Table 10, where vision-only receives 47.37 nDCG@10 while the other modalities are in the 60s range. This indicates that when a query has a clear signal in one of these textual modalities, the model is highly effective at retrieving relevant documents. Queries categorized as 'Base'—which may rely more on holistic video understanding or a combination of visual information and less distinct textual cues—exhibit a comparatively lower performance.

Table 4: Results on Video-MME and LongVideoBench with different frame retrievers.

| Frame Retriever | Modality | # Frames | LongVideoBench | Video-MME w/o subs | Video-MME w/ subs |
|---|---|---|---|---|---|
| No Sample | - | 768 | 55.67 | 53.10 | 62.30 |
| Uniform Sample | - | 100 | 52.30 | 53.90 | 57.80 |
| LanguageBind | Vision | 100 | - | 53.60 | 57.30 |
| LanguageBind | Vision + Audio | 100 | 56.38 | 54.40 | 57.80 |
| CLAMR | Vision | 100 | - | 55.60 | 59.40 |
| CLAMR | Vision + Audio | 100 | **57.09** | **55.90** | **61.30** |

## 6.4 LONG VIDEO QA

**Setup.** To evaluate the effectiveness of CLAMR in a downstream scenario, we test on Long Video Question Answering (QA) tasks using two benchmarks: the long-video subset (30 - 60 minutes in length) of Video-MME (Fu et al., 2024) and the (900, 3600s] duration group from the dev set of LongVideoBench (Wu et al., 2024). Specifically, we set up a retrieval-augmented generation (RAG) pipeline: given a long video, the retriever first selects key frames relevant to the question, which are subsequently provided as input to a VLM (Qwen2.5-VL-7B-Inst) answerer. We compare CLAMR against several baselines: uniform sampling, and retrieval-based methods using LanguageBind (vision only and vision+speech modalities). LanguageBind was chosen as it is generally the second-best method in Tab. 1 on the averaged multimodal setting when taking both datasets into account. To isolate the contribution of retrieval quality, the answering model and input token budget are fixed across all methods. Each method retrieves exactly 100 frames, which are passed to the same VLM in the same format. We also include a no-sampling baseline, where we provide the whole video as input. For this baseline, we follow the official Qwen2.5-VL setting (Bai et al., 2025), which samples videos at 2 FPS and caps input at 768 frames per video, with the total number of video tokens not exceeding 24,576. For Video-MME, which optionally includes subtitle input, we evaluate both with and without subtitles. For LongVideoBench, since some queries are grounded in subtitle content, subtitles are always provided. Performance is evaluated in terms of QA accuracy.

**Results.** As shown in Tab. 4, CLAMR consistently outperforms all baseline retrievers across both datasets. On LongVideoBench, CLAMR achieves 57.09% accuracy, surpassing uniform sampling by 4.79%. On Video-MME, CLAMR outperforms LanguageBind by 1.50% without subtitles and 3.50% with subtitles. A paired significance test on per-example QA accuracy confirms these improvements are statistically significant ($p < 0.05$).

Overall, multimodal retrieval methods outperform single-modality ones, confirming that leveraging multiple sources (e.g., vision and audio) helps retrieve more relevant content. For example, even without subtitles, LanguageBind with both vision and audio inputs outperforms its vision-only variant. CLAMR outperforms the no-sampling baseline, which uses 768 frames. This result indicates that full-frame inputs often include irrelevant or distracting content, which can degrade answer accuracy (Wang et al., 2024). By contrast, CLAMR selects a compact, query-relevant subset of frames, promoting more focused reasoning and better QA performance.

## 7 CONCLUSION

We presented CLAMR, a novel contextualized late-interaction retriever for multimodal content retrieval that jointly encodes video frames, speech transcripts, on-screen text, and metadata within a unified vision-language backbone. To enable the model to dynamically select the most relevant modality for each query, we introduced MULTIVENT 2.0++, a large-scale synthetic dataset of modality-targeted queries built upon MULTIVENT 2.0, and a modality-aware contrastive training objective that explicitly guides the model to focus on the correct modality. Extensive experiments on both MULTIVENT 2.0++ and MSR-VTT demonstrate that CLAMR substantially outperforms strong single-modality and multi-modality baselines. Finally, we showed that CLAMR's improved retrieval translates to downstream benefits in long-video QA, where retrieval of a more focused, relevant frame set yields higher answer accuracy than uniform sampling or naive fusion strategies.

ETHICS STATEMENTS

We do not foresee any ethical implications beyond standard ethical and safety considerations that apply to AI research generally.

REPRODUCIBILITY STATEMENT

We provide the data and code in the supplementary. Details of the data and model implementations, as well as all hyperparameters can be found in Appendix A. All prompts used can be found in Appendix D.

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

## A  ADDITIONAL EXPERIMENTAL SETUP DETAILS

### A.1  CLAMR IMPLEMENTATION DETAILS.

We set the maximum query length to 64 tokens. Because queries are usually far shorter than the associated documents, we follow prior work (Khattab & Zaharia, 2020; Faysse et al., 2025) and mitigate this length asymmetry by appending placeholder tokens to each query. Specifically, we add five extra tokens to help with re-weighting the original query terms. For video input, we resize frames to $224 \times 224$ pixels and use the default processor to perform any extra transformations. The maximum token length for other textual modalities (ASR, OCR, and metadata) is set to 256. For

MULTIVENT 2.0++, we adopt the same modality configuration as MULTIVENT 2.0, relying on the pre-extracted features released by its authors. Concretely, each video contributes (i) up to ten key frames detected with PYSCENEDETECT[5], (ii) ASR transcripts generated by Whisper (Radford et al., 2022a), (iii) OCR using Etter et al. (2023), and (iv) textual metadata descriptions supplied with the dataset. For MSRVTT, we only extract ASR using Whisper V3.

**Omni-Model.** Processing audio consumes a significant number of tokens, which complicates training procedures requiring large batch sizes. Therefore, we limited audio input to a maximum of 30 seconds, corresponding to 750 tokens. For visual input, we uniformly sampled 10 frames. The maximum token length for OCR and metadata was set to 256. This configuration resulted in an average input length of approximately 2048 tokens per sample, enabling an effective batch size of 16 on four A100 80GB GPUs. We use the same settings for the other hyperparameters as CLAMR with Qwen-VL-2.5.

## B ADDITIONAL EXPERIMENTS

### B.1 EXPLORATION OF DIFFERENT CONTRASTIVE LOSS FORMULATIONS

We investigated two alternative formulations of the contrastive objective, each designed to progressively enforce the contribution of the single, most relevant modality signal.

**InfoNCE with Correct-Modality Positives.** To encourage the model to focus on the correct modality, we keep the same denominator but replace each positive with the score computed *only* on the correct modality $m_k^*$. The contrastive objective thus helps to also put the distance between the query and the document embedding that uses the correct modality closer:

$$\mathcal{L}_{\text{ModPos}} = -\frac{1}{b} \sum_{k=1}^{b} \log \frac{\exp\left(s_{k,k}^{m_k^*}/\tau\right)}{\exp\left(s_{k,k}^{m_k^*}/\tau\right) + \sum_{j=1, j\neq k}^{b} \exp\left(s_{k,j}/\tau\right)}. \tag{4}$$

**InfoNCE with Modality Negatives.** To comprehensively encourage the model to *distinguish* modalities, we treat (i) other documents, (ii) other modalities of the *same* document, and (iii) every modality of other documents as negatives. The loss becomes

$$\mathcal{L}_{\text{ModNeg}} = -\frac{1}{b} \sum_{k=1}^{b} \log \frac{\exp\left(s_{k,k}^{m_k^*}/\tau\right)}{\sum_{j=1}^{b} \sum_{m\in\mathcal{M}} \exp\left(s_{k,j}^{m}/\tau\right) + \sum_{j=1, j\neq k}^{b} \exp\left(s_{k,j}/\tau\right)}. \tag{5}$$

Together, the two objectives progressively strengthen the model's ability to attend to the correct modality.

**Results.** As detailed in Table 5, applying these additional constraints to the contrastive loss did not improve retrieval performance compared to our main CLAMR (row a). In fact, increasing the constraints led to a decrease in performance. CLAMR trained with correct-modality positives ($\mathcal{L}_{\text{ModPos}}$, row d) resulted in R@10 of 86.6 and nDCG@10 of 56.8. This is a decrease of 1.4 points in R@10 and 1.7 points in nDCG@10 compared to the baseline CLAMR (row a, R@10: 88.0, nDCG@10: 58.5). Employing the more stringent modality negatives ($\mathcal{L}_{\text{ModNeg}}$, row e) further reduced performance, with R@10 dropping to 84.7 and nDCG@10 to 54.8. This represents a decrease of 3.3 points in R@10 and 3.7 points in nDCG@10 relative to the baseline (row a). These findings suggest that the underlying assumption that a query is solely relevant to one specific modality might be overly restrictive. The retriever appears to benefit from leveraging contextual signals from all available input modalities rather than being forced to focus exclusively on a single "correct" one.

### B.2 ADDITIONAL VIDEO-TO-TEXT BENCHMARKS

We further evaluate on two widely used video-text retrieval benchmarks: DiDeMo and ActivityNet. The results, presented in Table 6, reinforce the findings from our paper. Our proposed method,

---

[5]https://www.scenedetect.com/

Table 5: Retrieval results on MULTIVENT 2.0++ with different training setups.

| | Method | R@1 | R@5 | R@10 | nDCG@10 |
|---|---|---|---|---|---|
| (a) | CLAMR w. Qwen-2.5-VL | **26.7** | **85.1** | **88.0** | **58.5** |
| (b) | Qwen-VL-2.5 + pooled representation | 21.6 | 74.8 | 81.6 | 52.2 |
| (c) | CLAMR w. Qwen-Omni | 25.5 | 81.1 | 85.2 | 55.7 |
| (d) | CLAMR w. $\mathcal{L}_{\text{ModPos}}$ | 25.0 | 82.4 | 86.6 | 56.8 |
| (e) | CLAMR w. $\mathcal{L}_{\text{ModNeg}}$ | 22.3 | 79.8 | 84.7 | 54.8 |

Table 6: Results on Didemo and ActivityNet.

| | **Didemo** | | **ActivityNet** | |
|---|---|---|---|---|
| Method | R@10 | nDCG@10 | R@10 | nDCG@10 |
| *Single-Modality* | | | | |
| Whisper + mCLIP | 2.69 | 1.54 | 5.96 | 3.29 |
| Video + mCLIP | 35.56 | 22.23 | 36.26 | 22.24 |
| Imagebind | 29.18 | 18.58 | 44.19 | 27.56 |
| LanguageBind | 45.72 | 30.53 | 56.01 | 37.04 |
| *Multi-Modality* | | | | |
| mCLIP (avg) | 31.47 | 19.44 | 33.9 | 19.71 |
| ImageBind (avg) | 13.05 | 8.22 | 10.53 | 5.32 |
| LanguageBind (avg) | 18.33 | 11.83 | 20.22 | 12.12 |
| *Ours* | | | | |
| CLaMR (VLM) | **49.80** | **32.14** | **58.17** | **39.24** |

Table 7: Efficiency analysis on contextualization.

| Model | Latency (s/example) | Max GPU Usage (MB) | TFLOPS |
|---|---|---|---|
| CLAMR w/o contextualization | 0.036 | 6288 | 57.77 |
| CLAMR w contextualization | 0.039 | 7460 | 62.30 |

CLaMR, consistently outperforms all baselines. Notably, naive fusion methods like averaging (avg) perform poorly, confirming our hypothesis that baseline models are "susceptible to noise" from less relevant modalities. CLaMR's late interaction mechanism successfully mitigates this issue.

## B.3 EFFICIENCY ANALYSIS

We analyzed the overhead of our proposed contextualization. We benchmarked the runtime, GPU memory usage, and FLOPs for CLAMR with and without contextualization on MULTIVENT 2.0++. The experiments were run on an instance with 8 NVIDIA A100 80GB GPUs and a batch size of 4 per device (effective batch size of 32).

As shown in Table 8, while joint encoding logically increases resource usage, the impact on inference latency is modest—an increase of only 0.003 seconds per example.

As shown below, while joint encoding logically increases resource usage, the impact on inference latency is modest—an increase of only 0.003 seconds per example. We believe this modest increase is a highly favorable trade-off for the substantial performance gains observed. As shown in Table 1, contextualization improves R@10 by 29.4% and nDCG@10 by 31.3%. Furthermore, it is critical to note that this encoding cost is primarily incurred offline during document indexing. The online retrieval latency, which depends on query encoding and similarity matching, is minimally affected, consistent with other late-interaction frameworks like ColBERT (Khattab & Zaharia, 2020).

Table 8: Efficiency analysis of different methods.

| Model | Latency (s/example) |
|---|---|
| mCLIP | 0.06 |
| ImageBind | 0.56 |
| LanguageBind | 0.21 |
| Qwen Pooled | 0.30 |
| CLAMR | 0.33 |

Table 9: Human and automatic validation of the queries generated for MULTIVENT 2.0++.

| Query Source | Human Validation (Binary %) | Automatic Validation (1-5 Scale) |
|---|---|---|
| Vision | 96.88 | 4.59 |
| Audio | 85.00 | 4.18 |
| OCR | 68.75 | 3.57 |
| Metadata | 95.00 | 4.55 |
| Average | 86.41 | 4.22 |

Second, we benchmarked CLAMR's end-to-end inference time against baselines. As there are different models and different inferences, we set the batch size to 1 and run on the same A100. The results in Table 8 show that CLAMR's latency is comparable to other large-scale models. This is particularly noteworthy given that CLAMR (3B parameters) is significantly larger than the CLIP-based models, yet offers far superior retrieval accuracy by processing multiple modalities. We will add this complete analysis to the manuscript to clarify the computational trade-offs.

We further break down the computation of CLAMR: The query encoding takes 0.31s, while the actual late-interaction similarity computation takes only 0.02s. This indicates that the interaction mechanism itself introduces negligible latency overhead compared to the necessary backbone encoding.

### B.4 QUALITY ASSURANCE OF MULTIVENT 2.0++ QUERIES

To ensure the quality and stylistic consistency of our synthetic queries, we employed an in-context learning approach using human-annotated examples from the original MULTIVENT dataset as prompts. We then validated the 372k generated queries using both manual and automatic assessments on sampled subsets.

**Human Validation.** We conducted a manual quality assessment on a random sample of 400 queries (from 100 documents). Two authors independently judged whether each query was relevant and logically derivable from its specified source modality. Disagreements were reconciled to reach a consensus. As shown in Table 9, this process confirmed a high average correctness of 86.41%. A qualitative review revealed that errors, particularly for OCR (68.75% accuracy), typically originate from noisy source data, such as nonsensical or empty transcripts from upstream OCR and ASR models. In contrast, queries derived from human-written sources (i.e., vision and metadata) demonstrated significantly higher quality.

**Automatic Validation.** To assess quality at a larger scale, we prompted *Gemma-3-27b-it* to rate the query-source relevance on a 1-5 scale for our 4,000-example validation set. The results align closely with our manual findings, yielding a high average score of 4.22 out of 5. Notably, the performance ranking across modalities was consistent with the human validation: Vision >Metadata >Audio >OCR.

### B.5 ABLATIONS ON MISSING MODALITY

We evaluated performance when only a single modality is present, or when one modality is removed. The results, compared to the full model (first row), are in Table 10.These results show that while

Table 10: Results on CLAMR with missing modalities on MULTIVENT 2.0++.

| | R@1 | R@5 | R@10 | nDCG@10 | nDCG@10 by query modality | | | |
|---|---|---|---|---|---|---|---|---|
| | | | | | Vision | OCR | ASR | Description |
| CLAMR (All Modalities) | 26.66 | 85.11 | 88.03 | 58.47 | 47.37 | 62.60 | 64.21 | 63.32 |
| Vision only | 14.63 | 52.79 | 60.17 | 37.05 | 33.17 | - | - | - |
| OCR only | 19.28 | 65.69 | 70.41 | 44.78 | - | 59.13 | - | - |
| Audio only | 18.68 | 66.56 | 70.94 | 45.47 | - | - | 60.77 | - |
| Metadata only | 19.81 | 69.68 | 73.87 | 47.74 | - | - | - | 59.86 |
| No Vision | 22.87 | 83.58 | 87.17 | 57.25 | - | 62.98 | 63.21 | 63.87 |
| No OCR | 24.87 | 84.11 | 87.37 | 57.56 | 47.72 | - | 65.53 | 62.26 |
| No Audio | 23.87 | 80.98 | 84.31 | 55.45 | 46.98 | 62.19 | - | 61.71 |
| No Metadata | 24.93 | 82.05 | 85.90 | 56.42 | 46.01 | 64.06 | 64.20 | - |

Table 11: Ablation study on MULTIVENT 2.0. B-C: impact of architectural and objective choices. D-H: CLAMR trained and tested on a single modality. I-L: same models tested with all modalities.

| | Method | Inference modality | R@1 | R@5 | R@10 | nDCG@10 |
|---|---|---|---|---|---|---|
| (A) | CLAMR | All | **26.66** | **85.11** | **88.03** | **58.47** |
| | *Architecture and training objective design* | | | | | |
| (B) | CLAMR without contextualization | All | 18.95 | 64.30 | 68.02 | 44.53 |
| (C) | CLAMR with $LI_{context}$ (instead of $LI_{mw}$) | All | 23.93 | 80.92 | 86.04 | 56.26 |
| | *Single-modality w. single-modality inference* | | | | | |
| (D) | CLAMR Vision | Vision | 16.22 | 57.58 | 65.49 | 40.71 |
| (F) | CLAMR Audio | Audio | 18.15 | 64.56 | 68.48 | 43.93 |
| (G) | CLAMR OCR | OCR | 19.68 | 62.10 | 67.95 | 43.19 |
| (H) | CLAMR Metadata | Metadata | 20.01 | 68.22 | 72.94 | 47.09 |
| | *Single-modality w. all-modality inference* | | | | | |
| (I) | CLAMR Vision | All | 23.93 | 76.06 | 82.78 | 53.62 |
| (J) | CLAMR Audio | All | 23.27 | 81.18 | 85.77 | 55.85 |
| (K) | CLAMR OCR | All | 24.40 | 82.38 | 86.37 | 56.97 |
| (L) | CLAMR Metadata | All | 22.27 | 80.92 | 85.84 | 55.60 |

CLaMR is most effective with full modality access, its performance degrades gracefully. Even with one modality removed, performance remains fairly strong, highlighting the benefit of our joint encoding approach.

To further understand this, we analyzed nDCG@10 broken down by query type. This analysis reveals the power of contextualization: performance on queries for a specific modality is often better when other modalities are present. For example, for OCR-targeted queries, performance is higher when all modalities except metadata are present (64.06) than with only OCR available (59.13). This demonstrates that the model effectively leverages context from other available sources.

## B.6 ADDITIONAL ABLATIONS

We compare the performance of models trained with only a single modality to those trained with multiple in Table 11. When restricted to their respective single modalities during inference, these models performed considerably worse than the full multimodal model. For instance, in row (D) CLAMR vision achieves a nDCG@10 of 40.71. Among these, the metadata modality proves to be the most informative single source, while video is the least informative. Interestingly, when these models are allowed to access all modalities during inference, their performance significantly improved. For example, CLAMR vision (I) with all modalities (i.e. not restricted to video at test-time), has its nDCG@10 from 40.71 to 53.62. This demonstrates the model's capability to leverage contextual information from auxiliary modalities at inference time, even if not explicitly trained on all of them simultaneously. We attribute this to two factors. First, our fine-tuning approach with LoRA preserves the core pre-trained VLM's ability to jointly encode multimodal inputs. Second, the late-interaction mechanism allows the model to dynamically select salient tokens from all available modalities at inference time, making it robust to noisy signals from modalities it was not explicitly fine-tuned on.

Table 12: Performance comparison of different training and inference objectives.

| Training Objective | Inference Objective | R@1 | R@5 | R@10 | nDCG@10 |
|---|---|---|---|---|---|
| $LI_{\text{context}}$ | $LI_{\text{context}}$ | 23.7 | 80.9 | 86.0 | 56.3 |
| $LI_{\text{mw}}$ | $LI_{\text{mw}}$ | 23.9 | 83.8 | 87.7 | 57.3 |
| $LI_{\text{context}}$ | $LI_{\text{mw}}$ | 24.4 | 79.7 | 84.2 | 54.7 |
| $LI_{\textbf{mw}}$ **(Ours)** | $LI_{\textbf{context}}$ **(Ours)** | **26.7** | **85.1** | **88.0** | **58.5** |

Table 13: Ablation on MSR-VTT.

| Method | R@1 | R@5 | R@10 | nDCG@10 |
|---|---|---|---|---|
| (A) CLAMR (Full) | **46.1** | **71.3** | **79.8** | **62.4** |
| (B) CLAMR w/o Context | 34.1 | 56.6 | 65.6 | 49.2 |
| (C) CLAMR w/ $LI_{\text{context}}$ only | 36.6 | 57.3 | 67.0 | 50.8 |

Table 14: Router experiment.

| Method | R@1 | R@5 | R@10 | nDCG@10 |
|---|---|---|---|---|
| Soft Router (Prob. Weighted) | 7.3 | 21.5 | 24.8 | 16.0 |
| Soft Router (Max Weighted) | 6.9 | 20.7 | 26.4 | 16.0 |
| Hard Router | 9.8 | 27.3 | 33.2 | 21.0 |
| **CLAMR** | **26.7** | **85.1** | **88.0** | **58.5** |

Despite these improvements, the performance of single-modality trained models still lags behind our full CLAMR (A), which was trained with all modalities. This is true even with inference across all modalities (I-L). For example, the best performing model in this category, CLAMR OCR with all-modality inference (K), achieves an R@1 of 24.40, which is 2.26 points lower than the full model's R@1 of 26.66. This indicates that while leveraging all modalities at inference is beneficial, training the model with comprehensive multimodal information leads to the most robust and effective retrieval system. The most significant performance decrease occurs when training exclusively on video, highlighting the crucial role of other modalities in multimodal video content retrieval. Training solely on visual information evidently leads the model to under-utilize these other important modalities.

**Late-interaction variants.** Empirically, we found that while $LI_{mw}$ is a superior training objective (as it teaches modality differentiation), $LI_{context}$ is more effective at inference because it allows the model to leverage corroborating signals across modalities. Table 12 illustrates this: utilizing $LI_{mw}$ for training and $LI_{context}$ for inference yields the best performance, confirming that dynamic cross-modal corroboration is crucial for final retrieval accuracy.

**Ablations on MSR-VTT.** We performed the same ablation study on MSR-VTT. Note that because MSR-VTT training data is single-modality (video/text), we cannot perform the single-modality training ablations (D-H), but we can test the architectural contributions (A-C). The results shown in Table 13 mirror our main findings: Contextualization and the $LI_{context}$ mechanism are critical for performance.

## B.7 ADDITIONAL ROUTING EXPERIMENT

We implemented a "Soft Router" baseline using a Gemma-1b-it classifier fine-tuned on the synthetic data to predict modality weights (softmax probabilities), which are then used to weight the retrieval scores. We show the result in Table 14. First, we found that predicting the correct modality is difficult even for the router (Accuracy 35% on synthetic data, 29% on human data), suggesting that looking at the query in isolation is insufficient. Second, as shown below, both soft and hard routing strategies underperform compared to the single best modality (LanguageBind) and significantly un-

| Query Token | Selected Token | Selected Modality |
|---|---|---|
| ***Ex 1: Correctly utilizing Metadata and ASR*** | | |
| Bennett | Bennett | Metadata |
| high | high | ASR |
| school | SCHOOL | OCR |
| wins | win | ASR |
| state | state | ASR |
| championship | title | Metadata |
| ***Ex 2: Handling Noisy OCR*** | | |
| Bailey | Bailey | Metadata |
| makes | Bailey | Metadata |
| speech | spoke | ASR |
| in | spoke | ASR |
| Springfield | Springfield | ASR |
| Illinois | Illinois | OCR |
| ***Ex 3: Handling Noisy OCR (Complex Split)*** | | |
| Shannon | en | OCR |
| Ros | Ros | OCR |
| sm | sm | OCR |
| iller | iller | OCR |
| infiltr | infiltr | Metadata |
| ating | infiltr | Metadata |
| al | al | OCR |
| -Q | -Q | OCR |
| a | a | OCR |
| ' | ' | OCR |
| eda | eda | OCR |
| chat | chat | OCR |
| rooms | rooms | Metadata |
| ***Ex 4: Utilizing different modalities*** | | |
| Bl | Bl | ASR |
| ount | ount | Metadata |
| stown | stown | Metadata |
| High | [Video] | Video |
| School | SCHOOL | OCR |
| Tigers | Tigers | ASR |
| football | Football | Metadata |
| preview | FOR | OCR |

Figure 4: Comparison of token alignment and modality prediction between $LI_{\text{context}}$ and $LI_{\text{mw}}$ across four query examples.

derperform CLAMR. This confirms that explicit routing often introduces noise, whereas CLAMR's implicit soft-weighting via late interaction is far more robust.

## C  QUALITATIVE EXAMPLES

We conducted a token-level alignment analysis on ground-truth positive query-video pairs. For each query token, we identify the specific document token that yields the maximum late-interaction score. Since our architecture explicitly tracks the source of every token during scoring, we can definitively state which modality the model "attends" to for each term. As shown in the examples below, the model successfully retrieves relevant terms by leveraging cross-modal context.

| Prompt Type | Prompt |
|---|---|
| **Filtering** | You are a helpful retriever. Given a query and a document, you need to determine if the document is relevant to the query. You only need to answer with 'yes' or 'no'.
Query: {query}
Document: {doc}
Answer: |
| **Generating** | Given four documents, generate a short query (less than 10 words) that is only related to the document {target_id}. The other three documents should not be related to the query.
Document 1: {doc_video}
Document 2: {doc_speech}
Document 3: {doc_ocr}
Document 4: {doc_description}
Query: |

Figure 5: Prompts used for filtering relevant modality and generating synthetic modality-specific queries.

| | |
|---|---|
| System Prompt: | You are an assistant that creates search queries that would help users find videos. Create a concise and specific query. Do not output any extra information. |
| User Message: | ## Examples

{ICL examples}

## Your Task

{Video data for this query type}
**Query:** |
| Video Examples: | **Video Title:** {Title}
**Query:** {Query} |
| ASR Examples: | **Video Speech:** {Speech}
**Query:** {Query} |
| OCR Examples: | **Video OCR:** {OCR}
**Query:** {Query} |
| Description Examples: | **Video Description:** {Description}
**Query:** {Query} |

Figure 6: Prompt structure for synthetic query generation for MULTIVENT 2.0++. The prompt begins with a system instruction, followed by a user message that incorporates in-context learning (ICL) examples and video data corresponding to one of the four specified modality types (Video Title, ASR, OCR, or Description).

## D PROMPTS

The prompts employed for generating synthetic training data for MULTIVENT 2.0++ are detailed in Figure 6. We also provide the prompt used for the router in Figure 7.

```
Prompt                  You are an expert query classifier.  Given
                        a user query, determine which modality
                        is most relevant for answering it.  The
                        possible modalities are:  video, speech, ocr,
                        description.  Respond with only the predicted
                        modality name.
                        Here are some examples:
                        {ICL Examples}
                        Now, classify the following query:
                        Query:  {Query}
                        Modality:
```

Figure 7: Prompt for router with GPT-4.1.

### D.1 SAFEGUARDS FOR MULTIVENT 2.0++

The videos utilized are from the MULTIVENT 2.0 dataset. We rely on the safeguarding measures implemented by the original authors for this content and do not redistribute the videos. For our synthetically generated queries, which were created using Gemma-3, our safeguarding strategy included: (1) Prompt Engineering: Prompts were designed to elicit factual, descriptive, and task-relevant queries suitable for video retrieval, thereby avoiding the generation of inappropriate outputs. (2) Limited Scope: The queries are specific to an academic video retrieval task, a characteristic that inherently curtails their potential for broader misuse.

## E LIMITATIONS AND BROADER IMPACT STATEMENT

This research introduces CLaMR, a multimodal retrieval model designed to dynamically leverage multiple content modalities (video frames, audio transcripts, OCR text, and metadata) to improve retrieval accuracy significantly. Given the broad applicability of such multimodal retrieval technologies, it has the potential for both positive and negative applications. In our work, we have taken in the design of the prompts to mitigate risk; however, like other retrieval methods, it could be applied in negative ways. In summary, we do not believe that our method has more potential for misuse or negative impact than any other retrieval method, and that its improvements offer substantial opportunities for positive use.

Our study addresses multimodal video retrieval, training the retriever with a contrastive objective that benefits from large batch sizes. GPU-memory limits confined us to a batch size of 16, and, in the Omni model, required shortening the context window for non-text modalities. We expect that techniques such as quantization, memory-efficient optimizers, and improved long-context handling will soon enable both larger models and substantially larger batches. Likewise, ongoing advances in late-interaction architectures and retrieval-system engineering should further boost accuracy while reducing latency.

We utilized the 3B parameter versions of Qwen-VL and Qwen-Omni due to computational constraints; training the 7B models would require significantly reduced batch sizes or context lengths, which would likely harm contrastive learning performance. However, based on trends in related vision-retrieval tasks (e.g., the Vidore leaderboard (Faysse et al., 2025)), retrieval performance consistently scales with model size. We expect that scaling CLaMR to 7B or larger backbones would yield further accuracy improvements, though this would come with increased indexing costs.

Regarding scalability: (1) The similarity calculation can be accelerated using approximate nearest neighbor search (e.g., Cho et al. (2024)) to minimize the number of documents requiring full late-interaction scoring. (2) While storage is higher, future work can leverage quantization to reduce the memory footprint. Given the substantial accuracy gains on complex queries, we believe this is a justifiable trade-off for high-precision applications.

## F    USE OF LLMS

We used LLMs for grammar correction and polishing our writing.

