# OpenReview forum: "CLaMR: Contextualized Late-Interaction for Multimodal Content Retrieval"
_ICLR.cc/2026/Conference — Submitted to ICLR 2026_

### Official Review · Reviewer_jDoX · 2025-10-26

**Soundness:** 3
**Presentation:** 3
**Contribution:** 3
**Rating:** 6
**Confidence:** 2

**Summary:**

The paper proposes CLAMR, a multimodal video retriever that jointly indexes video frames, speech, text, and metadata, dynamically selecting relevant modalities for queries. Unlike conventional systems that treat modalities independently, CLAMR uses a unified backbone for better contextualization. Trained on MULTIVENT 2.0++ with modality-targeted queries and a modality-aware contrastive loss, it outperforms existing retrievers in video content retrieval and long-video question answering.

**Strengths:**

1. The methodology of the paper is clearly written and easy to follow.

2. CLAMR achieves state-of-the-art performance on MULTIVENT 2.0++ and MSR-VTT.

3. Ablation studies demonstrate the contributions of different modalities to retrieval, showing that multimodal fusion significantly enhances performance.

4. The proposed contextualized late-interaction approach is simple yet effective, maintaining the efficiency of dual-encoder models while outperforming the averaging of modality-specific retrieval scores.

**Weaknesses:**

1. I don’t find major weaknesses in this paper.

**Questions:**

What is the influence of Qwen model size on multimodal retrieval?

---

> ### Author Response · Authors · 2025-11-20
> **Response to jDoX**
>
> We thank the reviewer for their positive assessment, noting our "simple yet effective" approach and "state-of-the-art performance."
>
> **\> Q1. Influence of Qwen model size.**
>
> We utilized the 3B parameter versions of Qwen-VL and Qwen-Omni due to computational constraints; training the 7B models would require significantly reduced batch sizes or context lengths, which would likely harm contrastive learning performance. However, based on trends in related vision-retrieval tasks (e.g., the Vidore leaderboard), retrieval performance consistently scales with model size. We expect that scaling CLAMR to 7B or larger backbones would yield further accuracy improvements, though this would come with increased indexing costs. We added a discussion on this size-accuracy trade-off in the revised manuscript.

---

> > ### Author Response · Authors · 2025-11-26
> >
> > This is a friendly reminder that the author-reviewer discussion period will end soon in 1 week (Dec 3rd 9pm UTC). We have uploaded our rebuttals and would be grateful if you could take a look.
> >
> > We are happy to answer any further questions you may have (or if your comments have been resolved, please feel free to revisit your ratings), thank you again!

---

> ### Comment · Reviewer_jDoX · 2025-11-28
>
> Thank you for the rebuttal. To the best of my knowledge, I have not identified any major weaknesses in this work. I will maintain my current score.

---

### Official Review · Reviewer_f3Zs · 2025-10-27

**Soundness:** 3
**Presentation:** 2
**Contribution:** 3
**Rating:** 6
**Confidence:** 4

**Summary:**

This paper addresses the problem of fusing information from various modalities for video retrieval. The authors propose CLAMR, a framework that applies a late-interaction mechanism, inspired by text-retrieval models like ColBERT, to a multimodal context encompassing video, audio, OCR, and metadata. The core idea is to jointly encode all modalities within a single vision-language model (VLM) to create contextualized representations, and then perform token-level similarity matching. To train the model to focus on relevant modalities, the paper introduces a modality-aware contrastive loss and a new large-scale synthetic dataset, MULTIVENT 2.0++, created using an LLM to generate modality-specific queries. The experiments show that CLAMR achieves strong performance, particularly on this new synthetic benchmark, outperforming several unimodal and multimodal baselines.

**Strengths:**

1. The paper tackles a well-recognized and challenging problem in multimodal retrieval—how to effectively combine signals from diverse and potentially noisy sources without performance degradation.

2. The application of late-interaction mechanisms from the text domain to a complex multimodal video scenario is a logical and interesting direction. It provides an alternative to more common early-fusion or simple late-fusion (score averaging) techniques.

3. The authors make a notable effort to address the scarcity of suitable training data by generating a large-scale synthetic dataset with modality-targeted queries. This resource could potentially benefit future research in the area.

**Weaknesses:**

1. The core technical contribution can be viewed as an application of the existing ColBERT architecture to a multimodal setting using a standard VLM backbone. While the engineering is non-trivial, the conceptual novelty is somewhat incremental, as it primarily combines and adapts existing components rather than introducing a fundamentally new retrieval paradigm.

2. The most impressive results (e.g., +25.6 nDCG@10) are reported on the authors' own synthetic dataset, MULTIVENT 2.0++. This raises significant concerns about evaluation validity. The model might be overfitting to the specific patterns, vocabulary, and artifacts of the LLM used for query generation, rather than learning a truly generalizable retrieval capability. The performance gains on the established, human-annotated MSR-VTT benchmark are far more modest, which may indicate that the practical impact on real-world queries is less significant than claimed.

3. The paper proposes a modality-aware loss (LImw) for training, which encourages the model to identify the single most relevant modality. However, at inference, a different, holistic scoring function (LIcontext) that aggregates scores across all modalities is used. This disconnect between the training objective and the inference procedure weakens the central claim of "dynamic modality selection." It's unclear whether the model is truly learning to select modalities or if the performance gain is simply a byproduct of a more complex training objective that acts as a form of regularization.

4.The late-interaction approach introduces significant computational and memory overhead during the offline indexing phase compared to standard dual-encoder models. Given that the performance improvement on MSR-VTT is not as dramatic as on the synthetic dataset, the practical utility of CLAMR is questionable for applications where efficiency is a key concern.

**Questions:**

1. How can you ensure that the substantial performance gains on MULTIVENT 2.0++ are not primarily due to the model learning the specific artifacts of the Gemma-3-27b-it query generator? Have you considered a cross-validation experiment where you train on queries generated by one LLM and test on queries generated by a completely different LLM?


2. Could you provide a clearer justification for the discrepancy between the training objective and the inference objective? If the goal is to teach the model modality selection, why not use a scoring function at inference that also reflects this selection process? Does this design choice not suggest that the model isn't actually performing explicit modality selection at test time?

3. The paper's baselines for multimodal fusion include simple averaging and a "hard" router. A potentially stronger and more relevant baseline would be a "soft-routing" or attention-based mechanism that learns to dynamically weight the similarity scores from different modalities based on the query. How would you expect CLAMR to perform against such a baseline?

4. Given the high indexing cost of late-interaction and the more moderate performance gains on MSR-VTT, what is the compelling practical argument for deploying CLAMR over a simpler, highly optimized dual-encoder model that is fine-tuned on the same comprehensive data?

---

> ### Author Response · Authors · 2025-11-20
> **Response to f3Zs (Part 1/2)**
>
> We thank the reviewer for the detailed review and for appreciating that our work tackles a "well-recognized and challenging problem" via a "logical and interesting direction." We value the recognition of our efforts to address data scarcity.
>
> **\> W1. Novelty.**
>
> While our architecture leverages the established ColBERT interaction mechanism, our core novelty lies in adapting Late Interaction (LI) to the multimodal video domain, which presents unique challenges regarding noise and modality selection that do not exist in text retrieval. Indeed, Reviewer cXTo highlights that our work "fill\[s\] the gap of late-interaction’s underutilization in multimodal video retrieval," and Reviewer jDoX considers the "contextualized late-interaction approach" as "simple yet effective" in "outperforming the averaging of modality-specific retrieval scores." We demonstrate that standard LI fails to capture these dynamics without our specific contributions: (1) Contextualization (Ablation B), (2) the Modality-Aware training objective ($LI\_{mw}$, Ablation C), and (3) the synthetic modality-targeted data (Ablations D-H). We provide the first empirical analysis of how LI functions (and where it fails) in multimodal settings.
>
> **\> W2. Evaluation validity and overfitting.**
>
> We wish to clarify a crucial point regarding our evaluation: The test set is not synthetic. While we train on the synthetic MULTIVENT 2.0++ (generated by Gemma), we evaluate on the human-annotated public test set of MULTIVENT 2.0 from Kriz et al., 2025 (Lines 278-280, 549). Therefore, the significant performance gains (+25.6 nDCG@10) cannot be attributed to overfitting LLM artifacts, as the model is being tested on human-written queries against real-world videos.
>
> Regarding MSR-VTT, the improvement (+5.9% R@1 over next best method) is statistically significant (Table 1). The gains are naturally more modest here because MSR-VTT is a video-only benchmark; it lacks the rich, auxiliary modalities (OCR, Metadata, diverse Audio) that CLAMR is specifically designed to exploit. MULTIVENT allows us to demonstrate performance in realistic, messy environments where multiple modalities *must* be used.
>
> **\> W3+Q2. Disconnect between training and inference objectives.**
>
> We address conflicts in two stages. First, during training, we use synthetic queries designed to be answerable by a single modality (Lines 262-263) combined with our Modality-Wise loss ($LI\_{mw}$), which forces the model to learn distinct modality representations. Second, during inference, we employ Contextualized Late-Interaction ($LI\_{context}$), which sums maximum similarities across all modalities (Eq. 1). This effectively **combines corroborating evidence from multiple modalities**, allowing the model to reinforce consistent signals while naturally filtering out conflicting or noisy information.
>
> Empirically, we found that while $LI\_{mw}$ is a superior training objective (as it teaches modality differentiation), $LI\_{context}$ is more effective at inference because it allows the model to leverage corroborating signals. The table below illustrates this: utilizing $LI\_{mw}$ for training and $LI\_{context}$ for inference yields the best performance.
>
> | Training Objective | Inference Objective | R@1 | R@5 | R@10 | nDCG@10 |
> | :---- | :---- | :---- | :---- | :---- | :---- |
> | $LI\_{context}$ | $LI\_{context}$ | 23.7 | 80.9 | 86.0 | 56.3 |
> | $LI\_{mw}$ | $LI\_{mw}$ | 23.9 | 83.8 | 87.7 | 57.3 |
> | $LI\_{context}$ | $LI\_{mw}$ | 24.4 | 79.7 | 84.2 | 54.7 |
> | **$LI\_{mw}$ (Ours)** | **$LI\_{context}$ (Ours)** | **26.7** | **85.1** | **88.0** | **58.5** |
>
> **\> W4. Inference Computational overhead vs. practicality.**
>
> We address efficiency concerns in Appendix B.3 (Table 7). CLAMR's total inference latency (0.33s/example) is comparable to a pooled embedding baseline (0.30s/example). We emphasize the breakdown of this cost: the query encoding takes 0.31s, while the actual late-interaction similarity computation takes only 0.02s. This indicates that the interaction mechanism itself introduces negligible latency overhead compared to the necessary backbone encoding.
>
> Regarding scalability: (1) The similarity calculation can be accelerated using approximate nearest neighbor search (e.g., Cho et al., 2024\) to minimize the number of documents requiring full late-interaction scoring. (2) While storage is higher, future work can leverage quantization to reduce the memory footprint. Given the substantial accuracy gains on complex queries, we believe this is a justifiable trade-off for high-precision applications.

---

> ### Author Response · Authors · 2025-11-20
> **Response to f3Zs (Part 2/2)**
>
> **\> Q1. Cross-validation and LLM artifacts.**
>
> As noted in W2, our evaluation is already a form of cross-validation: we train on Gemma-generated queries and test on human-generated queries. The strong performance confirms generalization. Furthermore, we verified that there is **no data leakage**: (1) The video split is strictly disjoint (training videos are never seen in testing), and (2) out of \~371k synthetic training queries, only two had exact string matches in the human test set (targeting different videos). This negligible overlap (\<0.001%) confirms that performance is driven by generalization, not memorization.
>
> **\> Q3. Soft-routing baseline.**
>
> We thank the reviewer for this excellent suggestion. We implemented a "Soft Router" baseline using a Gemma-1b-it classifier fine-tuned on the synthetic data to predict modality weights (softmax probabilities), which are then used to weight the retrieval scores.
>
> First, we found that predicting the correct modality is difficult even for the router (Accuracy \~35% on synthetic data, \~29% on human data), a phenomenon also discussed in \[1\], suggesting that looking at the query in isolation is insufficient. Second, as shown below, both soft and hard routing strategies underperform compared to CLAMR and also the single best modality (LanguageBind), as has been discussed in previous papers such as \[2\]. This confirms that explicit routing introduces bottleneck errors, whereas CLAMR's implicit soft-weighting via late interaction is far more robust.
>
> | Method | R@1 | R@5 | R@10 | nDCG@10 |
> | :---- | :---- | :---- | :---- | :---- |
> | Soft Router (Prob. Weighted) | 7.3 | 21.5 | 24.8 | 16.0 |
> | Soft Router (Max Weighted) | 6.9 | 20.7 | 26.4 | 16.0 |
> | Hard Router | 9.8 | 27.3 | 33.2 | 21.0 |
> | **CLAMR** | **26.7** | **85.1** | **88.0** | **58.5** |
>
> \[1\] Telling the What while Pointing to the Where: Multimodal Queries for Image Retrieval. ICCV 2021\.
>
> \[2\] What Makes Training Multi-Modal Classification Networks Hard?. CVPR 2020\.
>
> **\> Q4. Practical argument for CLAMR.**
>
> As detailed in W4, the online computational cost is minimal. The practical argument is the capability gap: standard dual-encoders fail to effectively integrate noisy, diverse modalities (as shown by the poor performance of "Avg" baselines (Ln 337-339 of Table 1). CLAMR enables effective retrieval in complex, real-world video scenarios where evidence is scattered across audio, text, and visuals—a capability that simple dual-encoders currently lack—with minimal online latency cost.

---

> > ### Comment · Reviewer_f3Zs · 2025-11-26
> > **To authors**
> >
> > Thank you for the author's reply. My problem has been solved, and I will keep my score.

---

> > > ### Author Response · Authors · 2025-11-26
> > >
> > > We are glad we resolved your problems -- we are happy to answer any follow up questions that allow you to revisit your score.

---

### Official Review · Reviewer_mR3J · 2025-10-31

**Soundness:** 2
**Presentation:** 2
**Contribution:** 2
**Rating:** 4
**Confidence:** 4

**Summary:**

This paper presents a contextualized late-interaction retriever for multimodal video content retrieval, designed to address limitations of conventional systems that treat modalities as independent or use naive fusion. The proposed model jointly encodes four modalities—video frames, transcribed speech, on-screen text, and metadata—via a unified vision-language backbone (or omni-model for raw audio) to enhance contextualization. Experiments show it outperforms single/multimodal baselines.

**Strengths:**

1. Unlike baselines that encode modalities separately, the proposed model uses a unified backbone for cross-modal contextualization.

2. The proposed MULTIVENT 2.0++ fills the gap of modality-specific training data, supporting effective modality selection learning.

3. The proposed model consistently outperforms baselines across MULTIVENT 2.0++, MSR-VTT.

**Weaknesses:**

1. The performance drops noticeably when vision is the sole relied-on modality, showing weaker handling of visual-only signals. Why is this? Why is "vision the least informative," as stated in Line 430? Could this be unfriendly to most scenarios (given that vision is the most common and readily available modality)?

2. Primary evaluations focus on MULTIVENT 2.0++ and MSRVTT; tests on other multimodal benchmarks (e.g., MSVD, DiDeMo, ActivityNet) are limited, reducing generalizability evidence.

3. Why was the ablation experiment performed on MULTIVENT 2.0 instead of MULTIVENT 2.0++? Furthermore, ablation experiments should also be performed on other datasets.

**Questions:**

Please refer to the weakness.

---

> ### Author Response · Authors · 2025-11-20
> **Response to mR3J**
>
> We thank the reviewer for their constructive feedback and for highlighting that our unified backbone and the MULTIVENT 2.0++ dataset successfully "fill the gap" in modality-specific training data.
>
> **\> W1. Vision is the least informative.**
>
> This observation is largely an artifact of the query construction. The "Vision" queries in our training set are derived from video titles (Figure 3), which are often short or abstract, whereas the test set consists of granular human-written queries. These "base" queries are created in this way because the video content is multilingual; lacking a strong multilingual VLM for generation, we resorted to using Gemma on the video titles. This distribution shift results in lower apparent performance for vision compared to modalities like OCR or Metadata, which have more direct textual grounding. We will rename this condition **"CLAMR Base"** in the paper to avoid confusion regarding the informative value of the visual signal itself.
>
> **\> W2. Tests on other multimodal benchmarks.**
>
> We have indeed evaluated CLAMR on other standard benchmarks. Results for DiDeMo and ActivityNet are provided in Appendix B.2 (Table 5). These experiments confirm the trends seen in the main paper: CLAMR consistently outperforms baselines (e.g., \+12.7 nDCG@10 on DiDeMo vs. mCLIP avg), and naive fusion methods often degrade performance. This reinforces the robustness of our method across different video domains.
>
> **W3. Ablation experiment datasets.**
>
> We apologize for the confusion regarding the dataset naming. The ablation studies were performed using models trained on MULTIVENT 2.0++ (synthetic) and evaluated on the **MULTIVENT 2.0 public evaluation set (human-annotated)**. We have clarified this in Section 6.2.
>
> To address your request for ablations on other datasets, we performed the same architectural ablation study on **MSR-VTT**. Note that because MSR-VTT training data is single-modality (video/text), we cannot perform the single-modality training ablations (D-H), but we can test the architectural contributions (A-C). The results below mirror our main findings: Contextualization and the $LI\_{context}$ mechanism are critical for performance.
>
> | Method | R@1 | R@5 | R@10 | nDCG@10 |
> | :---- | :---- | :---- | :---- | :---- |
> | (A) CLAMR (Full) | **46.1** | **71.3** | **79.8** | **62.4** |
> | (B) CLAMR w/o Context | 34.1 | 56.6 | 65.6 | 49.2 |
> | (C) CLAMR w/ $LI\_{context}$ only | 36.6 | 57.3 | 67.0 | 50.8 |
>
> We have updated the paper to include these experiments.

---

> > ### Author Response · Authors · 2025-11-26
> >
> > This is a friendly reminder that the author-reviewer discussion period will end soon in ~1 week (Dec 3rd 9pm UTC). We have uploaded our rebuttals and would be grateful if you could take a look. We have included the requested ablation studies on MSR-VTT (W3) and clarified that the lower vision performance stems from query construction artifacts rather than signal quality (W1).
> >
> > We are happy to answer any further questions you may have (or if your comments have been resolved, please feel free to revisit your ratings), thank you again!

---

### Official Review · Reviewer_cXTo · 2025-10-31

**Soundness:** 3
**Presentation:** 3
**Contribution:** 3
**Rating:** 4
**Confidence:** 4

**Summary:**

This paper presents CLAMR, a contextualized late-interaction retriever for multimodal video content retrieval, and makes three core contributions. Proposing a unified vision-language backbone that jointly encodes four modalities to enhance cross-modal contextualization, addressing the limitations of independent modality encoding in conventional methods. Introducing MULTIVENT 2.0++, a large-scale synthetic dataset with 371k modality-targeted queries, solving the scarcity of fine-grained modality-specific training data for multimodal retrieval.

**Strengths:**

The joint encoding of multiple modalities and modality-wise late-interaction balance fine-grained token-level matching and computational efficiency, filling the gap of late-interaction’s underutilization in multimodal video retrieval.
Evaluates on multiple benchmarks and conducts extensive ablations, ensuring the reliability of conclusions.

**Weaknesses:**

Focuses only on four modalities and does not explore other critical modalities in video content.
 Evaluations are primarily based on event-centric and general video datasets.
While the joint encoding backbone aims to align modalities, the paper lacks analysis of alignment failures in temporally or semantically misaligned video content.
The model’s performance heavily relies on high-quality ASR transcripts and OCR text, making it vulnerable to low-resource scenarios where these modalities are noisy or unavailable.
The paper does not provide interpretability into how the modality-aware mechanism selects the “most relevant” modality for a given query.

**Questions:**

How does the modality-wise late-interaction mechanism (LIₘᵥ) specifically resolve conflicts when multiple modalities contain relevant information? The paper emphasizes single-modality targeting but lacks analysis of cross-modal corroboration scenarios.
What are the root causes of the lower performance on vision-targeted queries?
For videos longer than 60 minutes, how does CLAMR’s frame sampling and retrieval efficiency degrade? Is there a strategy to optimize long-sequence processing?

---

> ### Author Response · Authors · 2025-11-20
> **Response to cXTo (Part 1/3)**
>
> We thank the reviewer for their thoughtful feedback and for recognizing that our approach "fills the gap" in multimodal video retrieval by balancing "fine-grained token-level matching and computational efficiency." We also appreciate your acknowledgment of our extensive ablations and the introduction of the MULTIVENT 2.0++ dataset.
>
> **\> W1. Other critical modalities in video content.**
>
> We aimed to be comprehensive by covering the primary modalities available in standard online video platforms (e.g., YouTube) that users typically query: visual frames, audio (speech/ambient), on-screen text (OCR), and metadata (titles/descriptions) (Lines 152-156). Could the reviewer clarify which specific additional modalities they believe are critical? We are happy to discuss how our architecture might extend to include them.
>
> **\> W2. Evaluations are primarily based on event-centric and general video datasets.**
>
> Our choice of datasets was driven by the specific requirements of the task. We prioritized MULTIVENT 2.0++ because it is uniquely suited for this problem—it is the only large-scale resource containing queries explicitly targeting different modalities. We included MSR-VTT as a standard, popular benchmark for general video retrieval. Additionally, we provide results on DiDeMo and ActivityNet in Appendix B.2, which further demonstrates the generalizability of our approach across different video domains.
>
> However, we acknowledge the importance of evaluating diverse video types. If the reviewer has specific non-event-centric datasets in mind that would strengthen our analysis, we would appreciate their suggestions and would be happy to discuss and try to include them.
>
> **\> W3. Analysis of alignment failures in temporally or semantically misaligned video content.**
>
> We acknowledge that temporal and semantic misalignment is a significant challenge, particularly given that MULTIVENT 2.0 includes videos in various languages (Line 23\) where ASR and OCR can be noisy (Lines 1188, 1396). Our model is implicitly designed to handle this "semantic misalignment" by learning to weigh corroborating evidence from clearer modalities while ignoring noise in others.
>
> We provide a detailed qualitative analysis in **W5**, but offer a key example here. In the query *"Bailey makes speech,"* we perform late-interaction via  $Li\_{mw}$ (Eq. 3), and the OCR modality is heavily misaligned/noisy (returning arbitrary characters like "Y" and "R"), and the model cannot retrieve the correct tokens, because it only selects OCR tokens. As we show in W5, CLAMR, which uses $LI\_{context}$ (Eq. 1), can successfully circumvent this problem by dynamically selecting tokens from different modalities.
>
> Example:
>
> **Query:** Bailey makes speech in Springfield Illinois
>
> | Query Token | Selected Token | Selected Modality |
> | :---- | :---- | :---- |
> | Bailey | Y | OCR |
> | makes | Y | OCR |
> | speech | R | OCR |
> | in | R | OCR |
> | Springfield | Springfield | OCR |
> | Illinois | Illinois | OCR |
>
> **\> W4. Reliance on ASR transcripts and OCR text.**
>
> While we use ASR and OCR as textual context, our architecture is not strictly bound to them. As noted in W3, the model learns to be robust to noise in these channels. Furthermore, we explicitly address the dependency on transcripts with our Omni-model variant (Section 3.2, Lines 187-190), which processes raw audio directly without requiring intermediate ASR. The Omni-model achieves similar performance, and on MSR-VTT, it outperforms the VLM variant on R@10 (Lines 396-397). We believe that while reliance on derived text will diminish as end-to-end multimodal models improve, the core retrieval challenge—learning to attend to the correct modality—remains constant.

---

> ### Author Response · Authors · 2025-11-20
> **Response to cXTo (Part 2/3)**
>
> **W5. Interpretability of the modality-aware mechanism.**
>
> We provide the query-specific analysis in Appendix B5 (Ln 359), analyzing the accuracy of the model in selecting the correct modality on a human-selected set of queries where relevance must come from only the specified modality. As shown in Table 9, CLAMR significantly improves accuracy, achieving an average of **76.4%** compared to the next best of 39.5%.
>
> To further demonstrate interpretability, we conducted a token-level alignment analysis on ground-truth positive query-video pairs. For each query token, we identify the specific document token that yields the maximum late-interaction score. Since our architecture explicitly tracks the source of every token during scoring, we can definitively state which modality the model "attends" to for each term. As shown in the examples below, the model successfully retrieves relevant terms by leveraging cross-modal context.
>
> **Example 1: Correctly utilizing Metadata and ASR**
>
> **Query:** Bennett high school wins state championship
>
> | Query Token | Selected Token | Selected Modality |
> | :---- | :---- | :---- |
> | Bennett | Bennett | Metadata |
> | high | high | ASR |
> | school | SCHOOL | OCR |
> | wins | win | ASR |
> | state | state | ASR |
> | championship | title | Metadata |
>
> **Example 2: Handling Noisy OCR**
> **Query:** Bailey makes speech in Springfield Illinois
>
> | Query Token | Selected Token | Selected Modality |
> | :---- | :---- | :---- |
> | Bailey | Bailey | Metadata |
> | makes | Bailey | Metadata |
> | speech | spoke | ASR |
> | in | spoke | ASR |
> | Springfield | Springfield | ASR |
> | Illinois | Illinois | OCR |
>
> **Example 3: Handling noisy OCR**
> **Query:** Shannon Rossmiller infiltrating al-Qa'eda chat rooms
>
> | Query Token | Selected Token | Selected Modality |
> | :---- | :---- | :---- |
> | Shannon | en | OCR |
> | Ros | Ros | OCR |
> | sm | sm | OCR |
> | iller | iller | OCR |
> | infiltr | infiltr | Metadata |
> | ating | infiltr | Metadata |
> | al | al | OCR |
> | \-Q | \-Q | OCR |
> | a | a | OCR |
> | ' | ' | OCR |
> | eda | eda | OCR |
> | chat | chat | OCR |
> | rooms | rooms | Metadata |
>
> **Example 4: Utilizing different modalities**
> **Query:** Blountstown High School Tigers football preview
>
> | Query Token | Selected Token | Selected Modality |
> | :---- | :---- | :---- |
> | Bl | Bl | ASR |
> | ount | ount | Metadata |
> | stown | stown | Metadata |
> | High | \[Video\] | Video |
> | School | SCHOOL | OCR |
> | Tigers | Tigers | ASR |
> | football | Football | Metadata |
> | preview | FOR | OCR |
>
> **\> Q1. How does the modality-wise late-interaction mechanism ($LI\_{mw}$) resolve conflicts?**
>
> We address conflicts in two stages. First, during training, we use synthetic queries designed to be answerable by a single modality (Lines 262-263) combined with our Modality-Wise loss ($LI\_{mw}$), which forces the model to learn distinct modality representations. Second, during inference, we employ Contextualized Late-Interaction ($LI\_{context}$), which sums maximum similarities across all modalities (Eq. 1). This effectively **combines corroborating evidence from multiple modalities**, allowing the model to reinforce consistent signals while naturally filtering out conflicting or noisy information.
>
> Empirically, we found that while $LI\_{mw}$ is a superior training objective (as it teaches modality differentiation), $LI\_{context}$ is more effective at inference because it allows the model to leverage corroborating signals. The table below illustrates this: utilizing $LI\_{mw}$ for training and $LI\_{context}$ for inference yields the best performance.
>
> | Training Objective | Inference Objective | R@1 | R@5 | R@10 | nDCG@10 |
> | :---- | :---- | :---- | :---- | :---- | :---- |
> | $LI\_{context}$ | $LI\_{context}$ | 23.7 | 80.9 | 86.0 | 56.3 |
> | $LI\_{mw}$ | $LI\_{mw}$ | 23.9 | 83.8 | 87.7 | 57.3 |
> | $LI\_{context}$ | $LI\_{mw}$ | 24.4 | 79.7 | 84.2 | 54.7 |
> | **$LI\_{mw}$ (Ours)** | **$LI\_{context}$ (Ours)** | **26.7** | **85.1** | **88.0** | **58.5** |
>
> **\> Q2. Root causes of lower performance on vision-targeted queries.**
>
> The lower performance on vision-targeted queries stems from a domain shift between training and testing. Our synthetic training data for the vision modality uses "Base" queries derived loosely from video titles (see Figure 3), whereas the test queries are written by humans who watched the video content. This discrepancy makes the vision evaluation harder than other modalities where the source text (e.g., OCR/ASR) is more directly aligned. To clarify this distinction, we will rename the "CLAMR Video" baseline to **"CLAMR Base"** in the final paper.

---

> ### Author Response · Authors · 2025-11-20
> **Response to cXTo (Part 3/3)**
>
> **\> Q3. Frame sampling and retrieval efficiency for long videos.**
>
> For the retrieval component, we adhere to the Qwen-Omni sampling strategy to leverage its pre-trained capabilities. Qwen2.5-Omni uses a dynamic frame rate synchronized with audio (interleaved 2-second chunks) and handles long videos via block-wise processing to decouple perception from modeling. It is trained on extended sequences up to 32k tokens to ensure robust long-context understanding.
>
> Regarding downstream utility, we demonstrate CLAMR's effectiveness on long sequences in Section 6.3. On the long-video subset of Video-MME (30–60 mins) and Long VideoBench (15–60 mins), CLAMR retrieves 100 relevant frames that significantly improve QA performance compared to LanguageBind (Table 3), validating its ability to process long contexts efficiently.

---

> > ### Author Response · Authors · 2025-11-26
> >
> > This is a friendly reminder that the author-reviewer discussion period will end soon in ~1 week (Dec 3rd 9pm UTC). We have uploaded our rebuttals and would be grateful if you could take a look. In our response, we provided a detailed qualitative analysis of modality selection (W5), clarified the distinction between our training and inference objectives (Q1), and addressed the performance on vision-targeted queries (Q2).
> > We are happy to answer any further questions you may have (or if your comments have been resolved, please feel free to revisit your ratings), thank you again!

---

### Author Response · Authors · 2025-12-02
**Summary of rebuttal for AC**

Dear AC,

To assist with your decision-making, we provide a brief summary of the updates made during the discussion period and the current status of reviewer engagement.

1. Summary of Work Performed During Rebuttal

In response to reviewer feedback, we conducted additional experiments and added significant analyses to the manuscript:
- **New Baseline (Soft Router):** Implemented a “Soft Router” baseline (Gemma-1b-it classifier) to demonstrate that our implicit late-interaction mechanism significantly outperforms explicit routing strategies (Requested by f3Zs).
New Ablations: Performed architectural ablation studies on the MSR-VTT dataset to prove the method’s robustness beyond our primary dataset (Requested by mR3J).
- **Qualitative Analysis:** Added detailed token-level visualizations and tables demonstrating how the model successfully handles noisy OCR/ASR by attending to corroborating modalities (Requested by cXTo).
- **Clarifications:** Explicitly clarified that our test set is human-annotated (not synthetic) and distinguished between “base” training queries and human test queries to explain vision performance (Addressing f3Zs, mR3J).

2. Summary of Discussion & Addressed Comments

- **Generalization (mR3J, cXTo):** We addressed comments about generalizability by providing results on DiDeMo and ActivityNet (Appendix B.2) and the new MSR-VTT ablations, showing consistent performance gains.
- **Vision Performance (mR3J, cXTo):** We clarified that lower vision-only scores result from a domain shift between synthetic training titles (base queries) and human-written test queries, rather than signal quality. We have renamed this condition “CLAMR Base” to avoid confusion.
- **Interpretability (cXTo):** We provided concrete examples of the model effectively filtering out noisy OCR (e.g., misrecognized text) by leveraging Metadata/ASR, validating the dynamic modality selection.
- **Evaluation Validity (f3Zs):** We resolved the comment regarding “overfitting to LLM artifacts” by confirming the evaluation is performed on a disjoint, human-annotated test set.
3. Reviewer Engagement & Status
- **f3Zs (Score: 6):** Actively engaged. Following our inclusion of the requested “Soft Router” baseline and clarifications on the test set, the reviewer stated: “My problem has been solved” and maintained their positive score.
- **jDoX (Score: 6):** Acknowledged the rebuttal and maintained their positive score.
- **mR3J (Score: 4) & cXTo (Score: 4):** We have uploaded the requested ablations (MSR-VTT) and qualitative analyses (interpretability) but have not yet received a follow-up response.

We believe we have objectively addressed the core technical questions raised by all reviewers.

---

### Meta-Review · Area_Chair_1Auw · 2025-12-07

**Summary:**

This paper addresses the problem of fusing information from various modalities for video retrieval. The authors propose CLAMR, a framework that applies a late-interaction mechanism, inspired by text-retrieval models like ColBERT, to a multimodal context encompassing video, audio, OCR, and metadata. The core idea is to jointly encode all modalities within a single vision-language model (VLM) to create contextualized representations, and then perform token-level similarity matching. To train the model to focus on relevant modalities, the paper introduces a modality-aware contrastive loss and a new large-scale synthetic dataset, MULTIVENT 2.0++, created using an LLM to generate modality-specific queries. The experiments show that CLAMR achieves strong performance, particularly on this new synthetic benchmark, outperforming several unimodal and multimodal baselines.

At the initial rating stage, two reviewers are positive while the other two reviewers are negative. The main concerns are with the experiment part, including the evaluation on more general datasets, reliance on ASR transcripts and OCR text, and insufficient ablations on different datasets. Although the authors have partially addressed the issues, the AC thinks this paper still requires a major revision before it can be accepted for a top-tier conference. Therefore, the current recommendation is Reject.

**Reviewer Concerns:**

The authors have addressed some concerns, such as:
1. Analysis of alignment failures in temporally or semantically misaligned video content.
2. Interpretability of the modality-aware mechanism.
3. Clarification on the novelty.
4. Disconnect between training and inference objectives.
5. Baselines and some detailed illustrations.

However, the experiments are insufficient and not convincing:
1. Lack of evaluation on more general and real-world datasets,
2. Reliance on ASR transcripts and OCR text.
3. Insufficient ablations on different datasets.

**Reviewer Scores:**

The AC thinks the negative reviewers will maintain their scores.

---

### Decision · Program_Chairs · 2026-01-26

Reject